# HESS Opinions: Repeatable research: what hydrologists can learn from the Duke cancer research scandal

Michael N. Fienen[1] and Mark Bakker[2]

[1]U.S Geological Survey Wisconsin Water Science Center, Middleton, Wisconsin, USA
[2]Water Resources Section, Faculty of Civil Engineering and Geosciences, Delft University of Technology, Delft, Netherlands
*Correspondence to:* Michael N. Fienen (mnfienen@usgs.gov)

**Abstract.** In the past decade, difficulties encountered in reproducing the results of a cancer study at Duke University resulted in a scandal and an investigation which concluded that tools used for data management, analysis, and modeling were inappropriate for the documentation of the study, let alone the reproduction of the results. New protocols were developed which require that data analysis and modeling be carried out with scripts that can be used to reproduce the results and are a record of all decisions and interpretations made during an analysis or a modeling effort. In the hydrological sciences, we face similar challenges and need to develop similar standards for transparency and repeatability of results. A promising route is to start making use of open source languages (such as R and Python) to write scripts and to use collaborative coding environments (such as Git) to share our codes for inspection and use by the hydrological community. An important side-benefit to adopting such protocols is consistency and efficiency among collaborators.

## 1 Introduction

In hydrology, we face increasing amounts of data that we use to build and calibrate models, which are ultimately used for forecasts. Many subjective and interpretive steps go into the translation of data to models, sometimes referred to as the "art of hydrology" (Savenije, 2009). Hydrological science always involves judgements and interpretations so it is unrealistic to expect a single path from original data to models (Fienen, 2013). However, we can certainly do a better job of documenting our interpretations, and make it easier for others to repeat, if not reproduce, our results. The field of cancer research faced a scandal in the past decade, related to applications of omics, that offers lessons for hydrology both in the nature of the scandal and in the response by institutions involved in and overseeing cancer research.

In this Opinion Paper, we provide background about the Duke cancer scandal, highlight how repeatability and reproducibility were at the center of the solutions, and relate lessons from the scandal to the field of hydrology. Unfortunately, other high-profile scientific scandals have taken place—sometimes due to neglect, and sometimes due to intentional fraud—but we focus on the Duke cancer scandal to highlight requirements that came out of the scandal which have relevance to hydrology.

## 2 The Duke Cancer Scandal

In 2007, a comment on a paper in Nature Medicine pointed out difficulties in reproducing a cancer study at Duke University in the research group of Anil Potti (Coombes et al., 2007). This spiraled into "the Duke cancer scandal" that included allegations of improper methods and inflated credentials. The scandal led to an internal inquiry (Califf and Kornbluth, 2012) and later a set of guidelines by the Institute of Medicine (Institute of Medicine, 2012) highlighting the shortcomings of the studies and putting forth protocols to avoid such problems in the future. A key element of the guidelines was that an unreproducible path through data using graphical user interfaces, spreadsheets, and other such tools would no longer suffice to document the data management that necessarily precedes analysis and modeling. Computations should be "locked down" and repeatable using scripting languages so that, given an original set of data, all steps of analysis can be repeated and documented (Institute of Medicine, 2012).

The field of omics in which the Potti group performed research refers to fields in life sciences ending in "-omics", and is defined as "...the scientific disciplines comprising the study of global sets of biological molecules such as DNAs (genomics), RNAs (transcriptomics), proteins (proteomics), and metabolites (metabolomics)..." (Carlson, 2012). Omics is a powerful field with many applications in the life sciences including enabling cancer researchers to use large datasets to explore the efficacy of cancer treatments based on patient data and statistical modeling prior to conducting trials in humans. The large datasets require processing to remove unsuitable data for a particular experiment. However, if too many data are removed in the process, overfitting can result "which unintentionally exploits characteristics of the data that are due to noise, experimental artifacts, or other chance effects not shared between data sets rather than to the underlying biology" (Carlson, 2012). As a result, the provenance of the data ultimately used for experiments is a critical element to the overall work, and the analysis path can be tedious and involve subjective judgement, especially with large, complicated datasets. Indeed, "guaranteeing robust data provenance and reproducible data management" (Califf and Kornbluth, 2012) was cited as a major recommendation by the Duke University internal inquiry. Key elements were to establish data provenance are the use of scripting languages and the sharing of code (Califf and Kornbluth, 2012).

## 3 Reproducible or Repeatable?

The National Institute of Standards and Technology in the USA defines "reproducible" as "closeness of the agreement between the results of measurements of the same measurand carried out under *changed* conditions of measurement" and repeatability as "closeness of the agreement between the results of successive measurements of the same measurand carried out under *the same* conditions of measurement" (Taylor and Kuyatt, 1994). These definitions are very similar, but the subtle distinction (highlighted in italics) is important. For a process to be reproducible, it implies that a different group given the same data and following the same protocols will interpret and process them the same way, resulting in the same outcome as another group.

On the other hand, a repeatable process is one in which all steps are documented and the exact steps of data processing can be repeated. In fields such as omics and hydrology, where judgement and interpretation are part of the process, the goal is often

more repeatability than reproducibility. For a repeatable path through the data, with judgements properly documented, another research group can evaluate each judgement and decide whether to agree with it or not.

The call for repeatable research has echoed through the computational sciences for several decades (Fomel and Claerbout, 2009), although the terms reproducible and repeatable are often used interchangeably. Peng (2011) presents a spectrum of reproducibility from solely publication of results (not reproducible) to inclusion of code, code plus data, or linked and executable code and data (full reproducibility, which should probably be called repeatability). Some journals have adopted policies to encourage repeatability of results, varying from a requirement to state where or how the data can be obtained to the submission of code that can be run to actually repeat the results, including "kite marks" that indicate which level of repeatability/reproducibility a paper achieves (Peng, 2011).

Reproducibility may be seen as a higher goal than repeatabilty. Unfortunately, hydrological field experiments are typically not made under controlled conditions such as bench experiments in chemistry or physics, but rather depend on natural variability in conditions like precipitation, river stage, and others, which may make reproducibility an elusive goal. Furthermore, many quantities are measured only indirectly and strongly depend on interpretation and inverse modeling, including remotely sensing and geophysical imaging. Other data sources are less quantitative but more descriptive, such as land use, boring logs, and outcrop analysis. Given the uncertain nature of all these data sources, it is understandable that conclusions drawn from hydrological models can be highly uncertain. Quantification of the uncertainty and problems of equifinality are very important and beyond the scope of this Opinion Paper, but they are certainly not an excuse to play down the importance of repeatability. On the contrary, repeatability seems to be the first step to tackle the problem of uncertainty and equifinality.

## 4   How does this relate to hydrology?

The fields of omics—as used in cancer research—and hydrology may seem as completely unrelated, but the way data are handled and processed, and the ramifications of such data handling are actually quite similar. Hydrological and omics datasets can both be noisy and require trimming or even adjustment of some values based on quality control, interpretation, and appropriateness for the analysis at hand. Hydrological datasets come in an incredible variety of data types and formats, such as meteorological data, water levels, flow measurements, soil types, lithological logs, surface water diversions, groundwater extractions, and remote sensing data. Much of this information is provided in spreadsheets, graphical documents, databases, and web-queries. At the raw data stage, the provenance is generally known but between data acquisition and creating model inputs and outputs, an unknown series of steps takes place that breaks the provenance and can hide the interpretations and judgements that took place.

Beyond interpreting the same spreadsheets and databases, many hydrologists use graphical user interfaces (GUIs) to organize and manipulate the information used in models. In a GUI, data are interpreted spatially and temporally, boundary conditions are specified, grids are generated, parameters are selected or specified, etc., while typically none of these steps can be repeated without going through the same sequence of mouse clicks, menu selections, and entries made in boxes. Repeating all these steps is tedious, prone to errors, and does not include documentation of interpretations made.

As time passes after the completion of a modeling or analysis project, the collection and interpretation of the original data is often of more lasting use than the actual model files. Modeling technology changes but the data are persistent. Access to the original data and a detailed documentation of the analysis path may be the most useful record of a project in the future (e.g. Anderson et al., 2015).

## 5    What can be done?

In the same spirit as the recommendations of the Institute of Medicine report above, scripting languages such as R and Python can replace much of the GUI and spreadsheet data manipulations in hydrology and hydrological modeling. Scripting languages have many features and access to specialized libraries. They also have facilities for making comments in which the subjective elements of data processing can be clearly stated. In this way, common tasks (e.g., unit conversions), specific decisions (e.g., identification of outliers), and algorithms (e.g., spatial interpolation or regularization of time intervals) can be reviewed and understood. Scripting languages are interpreted so they do not need to be compiled, making them work on many different platforms easily. Tools like Jupyter Notebooks (formerly IPython Notebooks; Pérez and Granger, 2007) and RStudio (RStudio Team, 2015) provide seamless integration of written documentation and executable code. In addition to repeatability, an important benefit of these tools is increased efficiency. Note that several Python packages are specifically designed for hydrologists, for example for watershed modeling (Lampert and Wu, 2015) and groundwater modeling (Bakker, 2013; Bakker et al., 2016).

Of course, this implies that everything can be done without a GUI, but that is not necessarily true. GIS software and model GUIs provide a valuable set of tools to enable model creation and data analysis. We suggest, however, that an auditable scripting path through the GUI logic is a necessary feature of a GUI to record the many steps taken in the model-building process. For example, ArcGIS (ESRI, 2011) provides a Python application programming interface that can be used to perform any operation using a script. Furthermore, it is possible to record all the steps while clicking and selecting in the GUI as a Python script that serves as a record of the performed analysis that can be evaluated and run later, mitigating the hurdle of programming expertise for practitioners to improve repeatability in their work.

## 6    What else can be done?

In hydrological modeling, the documentation of a data analysis and modeling effort in a script is only one side of the coin. The other side of the coin is the model that is used to perform the computations. Without the availability of an executable code, the simulations can still not be repeated and without the availability of the code itself, the computational steps in the code cannot be understood and scrutinized. The code is also necessary to run the program on another platform than the authors used or a future version of the same platform. Harvey and Han (2002) already recognized the increasing value of open-source codes in hydroinformatics. Ince et al. (2012) make a strong case that "anything less than the release of source programs is intolerable for results that depend on computation".

Over the past decade and a half, open-source codes have risen in prominence, as illustrated in an analysis of data analytics job postings in 2015, showing more requests for open-source coding experience than experience with proprietary analytics codes (http://r4stats.com/articles/popularity/). Unfortunately, many research groups don't make/have time to go through the extra effort to extensively test and document their code and make it available to the public. Merali (2010) suggests that more open-source software may be developed at universities when the value of such developments are rewarded more appropriately (e.g., similar to research papers in peer-reviewed journals). The road for sharing computational codes is paved by the emergence collaborative coding environments such as Git (Chacon, 2009), an easy to use and free application for version control of (collaborative) coding efforts, the success of github.com, bitbucket.org, and other free hosting services for the dissemination of source code, and the availability of free and open-source compilers for many languages.

It is noted that open-source software is not always free and the open-source aspect of the code is not a panacea—indeed, proprietary software may also be used to improve repeatability. However, the more open all aspects of the analysis are, the more transparent are the findings. Both open-source and proprietary software used to enhance repeatability and transparency should be documented in enough detail to allow benchmarking and comparisons by the community to ensure consistency between documented processes and their outputs.

## 7 Conclusion

This paper began with a short review of a cancer scandal, which started when difficulties were encountered in the reproduction of a cancer study at Duke University. On the face of it, the fields of hydrology and omics may seem unrelated. However, both fields need to make important forecasts, whether it is the response of patients to cancer treatment, high water levels in rivers, droughts, or contaminant plume migration in groundwater systems. Both fields depend on drawing conclusions from models based on large datasets. In both fields, processing, trimming, and validating these datasets require judgement and a certain degree of art and interpretation. The specific interpretations and decisions can make the difference between high-quality forecasting and overfitting where the model chases noise in the dataset at the expense of generalization. Uncertainty in the entire data analysis process contributes to nonunique solutions in modeling and analysis. It is crucial to understand all decisions made in research that lead to a conditionally unique solution or an ensemble of solutions.

For decades, both omics and hydrology have seen a variety of techniques for data analysis and interpretation, including GUIs, custom programs, manipulation of spreadsheets, and hand calculations. GUIs and spreadsheets typically do not provide an auditable path through the process and some custom programs, once compiled, are opaque to review if source code is not provided. The result is a lack of transparency and repeatability that may cover-up mistakes, judgements based on thinking that can change over time, and, at worst, manipulation or fraud.

The cancer research problems were encountered when one group tried to confirm the analysis and modeling of another group—a scientific tradition that is not conducted frequently in the hydrological sciences. During the investigation of the Duke cancer scandal, it became apparent that mistakes of overfitting were made. The response of academia and the Institute of Medicine was to require data provenance and documentation of data processing and modeling in scripts such that all steps

could be repeated independently and the analysis path through the data was well documented. These new requirements caused a major shift in approach for many researchers. The field of hydrology has not experienced such a high-profile scandal, but we must learn preemptively and adopt similar standards of transparency and repeatability for our work. Scripting languages (such as R and Python) and collaborative coding environments (such as Git and online hosting such as github.com and bitbucket.org)

5  make it practical to improve the repeatability and documentation of our research. Furthermore, transparency and reproducibility are enhanced by the application of open-source software.

Open data are also the subject of an initiative in the US at the direction of the White House (https://www.whitehouse.gov/sites/default/files/omb/memoranda/2013/m-13-13.pdf). This initiative has created an environment in which researchers employed by the US government must now adhere to much higher standards of repeatability and

10  data stewardship (similar open data initiatives are explored by the Horizon 2020 research program of the EU). Such requirements come at a cost of time and energy. To make it more realistic for such standards to be adopted, the academic systems of rewards must evolve to properly reward the extra effort required. It is up to individual scientists, journals, stakeholders, and funding agencies to demand it and create meaningful standards of repeatability.

It is not fully necessary to hold all research to exactly the same standard, but if we, as a community, assign value to repeata-

15  bility and transparency, then even voluntary standards can gain currency. The entire community can benefit from the ability to build on each others' prior work when both data and auditable code are available. Important advances in science are made when results are confirmed or falsified in subsequent research. In any case, we must learn from the Duke cancer research scandal to prevent our field of hydrology from falling in the same trap.

**Disclaimer**

20  Any use of trade, product, or firm names is for descriptive purposes only and does not imply endorsement by the U.S. Government.

*Acknowledgements.* The authors thank Chandra Miller Fienen (formerly BIOARRAY Therapeutics) for informing us about the Duke cancer research scandal, Yu-Feng Lin (Illinois State Geological Survey), and Randall Hunt (U.S. Geological Survey) for valuable conversations on the topic, and the online reviewers for participating in a robust discussion about the manuscript.

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
