# Peer review of "HESS Opinions: Repeatable research: what hydrologists can learn from the Duke cancer research scandal"

_Hydrology and Earth System Sciences, 2016_

## Referee Comment (RC1) · O. A. Cirpka (Referee) · 26 May 2016

In this opinion contribution, the authors take a recent scandal in cancer research, where the processing of molecular-biological data turned out to be so intransparent that the analysis could not be repeated, as an opportunity to call for equally strict rules of conduct in hydrological data processing as has now been called upon in the wake of the mentioned scandal in cancer research. Being an opinion paper, the paper need not be balanced, however, I believe that the authors should be advised to (a) discuss counter-arguments and obstacles, (b) rethink whether repeatability is really more important than reproducibility, (c) broaden their perspective to reach out to hydrological researchers who are not modelers, and (d) separate issues of transparency from the

call of open-source programing.

On page 2, lines 9-11, the authors state: "Omics is a powerful field enabling cancer researchers to use large datasets to explore the efficacy of cancer treatments based on patient data and statistical modeling prior to conducting trials in humans." While this is not wrong, it gives the erroneous impression that omics (summarizing (meta)-genomics, measuring DNA, (meta)-transcriptomics, measuring messenger RNA, pro-teomics, measuring proteins, and metabolomics, measuring metabolites) was specific to cancer research. This is not the case. The mentioned molecular-biological techniques dominate all modern life sciences, including environmental microbiology, where it actually may come into contact with hydrological research. Modern omics-technologies, such as quantitative polymerase chain reaction (qPCR) as example for genomics, make it possible to test for 1000 genes with a single test. In chromatography coupled to high-resolution mass-spectroscopy, as used for metabolomics studies, thousands of molecular features are measured in a single sample. Analysis of some omics data is not possible without bioinformatics tools, operated by computer scientists rather than biologists and medical researchers.

Just to put this into its own context: Modern life sciences would be impossible without these techniques, and they are accompanied by other data-intensive technologies, such as practically all imaging techniques. Medical imaging and geophysical surveying are based on essentially the same measurement principles, some of which producing gigabytes of raw data. Life scientists are typically not considering the raw data at all and work with the images instead, which are inversion results prone to inversion arti-facts, and which are often "enhanced" to highlight the features of interest (such as a brain tumor in an MRI scan).

The authors claim that hydrologists are more or less in the same situation. I would like to question that. Most hydrologists have a strong quantitative physical background. I would not take a hydrologist seriously, who takes a satellite-based soil-moisture map for granted. Everybody in the community knows that this is an interpreted satellite product

rather than a direct measurement of volumetric water content. In general, most hydrologists are closer to the physics of the measurements and more aware of the pitfalls of their quantitative interpretation than most life-scientists. In physical sciences, a measurement without an error analysis is considered useless, which is an attitude often lacking in life sciences and biogeochemistry. As such, there may be better chances to call for repeatable research, in the sense defined by the authors, in hydrology than in life sciences because the raw data are closer to our own way of thinking. This does not mean that we are better people or better researchers; it may actually mean that most our measurements are still so primitive that we ourselves can directly understand them.

A second very important set of differences between research on human health and hydrology is the question what is at stake and what are the commercial interests. Wrong interpretations by human-health researchers may eventually cost human lifes in a fairly direct sense. Despite all the yada yada regarding the importance of hydrology for human well-being, erroneous smoothing of a hydrograph most likely will not have the same direct effect. Conversely, there is also much less pressure on hydrologists to give the interpretation of their data a certain twist, because there is no equivalent to the pharmaceutical industry investing hundreds of million dollars to develop a drug. There are good reasons why the US Food and Drug Administration (FDA) has defined extremely strict codes on planning and documenting every decision made in developing, manufacturing, and distributing drugs. But it comes at the price of endless red tape, which honestly I don't want to see in hydrological research. For that I am freely willing to confess that my research is not directly saving lifes.

There are of course hydrology-related studies supporting high-risk decisions of high societal relevance where there is lots at stake. Analyzing potential sites for a nuclear-waste repository would fall into that category. Following an FDA-like protocol in such applications is highly recommendable, and most certainly done, simply because any decision taken will sooner or later end up in court anyway. Most hydrological studies,

however, don't fall into that category. This allows us to be somewhat more relaxed.

The authors try to work out differences between reproducibility and repeatability and make the bold statement that repeatability is more important. I like to doubt that. While the difference appears quite semantic to begin with, the authors highlight what they want: Given the same data, everybody should be able to follow the same steps of data processing to come to the same conclusions. But is lacking repeatability in this sense really the biggest problem?

I would claim that hydrologists can learn a lot from life scientists regarding the importance of reproducibility rather than repeatability. Andrew Binley tried to explain to hydrogeophysicists at a recent AGU fall meeting what he has learned in a collaboration with crop scientists where they used geophysical methods to test whether certain crop genotypes are more efficient in their water use than others, which involved geophysical imaging of soil moisture (Shanahan et al., 2015). While geophysicists (like hydrologists) like to get the last piece of information out of a single measurement campaign, life scientists (including cancer researchers, crop scientists, and ecologists) don't trust single experiments at all. You need several plots of the same experiment, you need several sites, and in a comparison study you have to make sure that either all other factors influencing the outcome of your experiment are identical or that the sample size is large enough for randomization, before you draw any conclusions. All of this has to do with reproducibility of experiments rather than data processing. If you cannot confirm your findings in a repeated experiment, you may have interpreted a freak event. Biologists take extra classes on experimental design, which discuss the statistical basics of reproducible research of complex systems. By contrast, there is a tremendous tendency in physical environmental sciences (including hydrology) to over-interpret single experiments. I don't think that coming to the same conclusions if you re-analyze the same data set is sufficient at all. Frankly, I also don't think that the latter has not happened in hydrology: There are dozens of papers on analyzing the same tracer tests in the aquifers of Borden, Cape Cod, and at the MADE site in Columbus, Mississippi. A

single experiment on mixing-controlled reactions on the bench-scale (Gramling et al., 2002) has been re-interpreted over an over again, whereas hardly any attempts were made to repeat the actual experiment.

So quite in contrast to the authors, I believe that reproducibility is a major concern in physical hydrology, which actually has consequences on the way how we do research: Should we intensify and refine measurements (and their associated interpretations) at single sites to gain the most mechanistic system understanding, or should we spread out cheaper measurements over many sites to gain reproducibility and confirmation of findings, possibly at the cost of refraining from a fully mechanistic process description? In inter-disciplinary research collaborations, these questions are heatedly debated, and putting down the necessity of reproducibility does not truly help.

I agree with the authors that transparency in data processing is mandatory. But this may be achieved by other measures than enforcing everybody to use open-source codes written in a free-of-charge scripting language. I want to highlight that transparency should start with the selection of the data. As a frequent reviewer, I am often bored by reading long explanations that the time series 3, 7, and 11 out of 15 time series measured were excluded for lengthy explained reasons, which is followed by an extended disucssion of outliers. However, much worse would be dropping the three time series and all outliers altogether and pretending that only 12 surprisingly consistent time series were taken. The same holds for measuring multiple chemical species but discussing only selected ones without mentioning the selection at all. We honestly don't know how much data has been droppped in hydrological studies, and how often this was justified (noisy probe, no flow through the piezometer, no gain of information at all, etc.) versus how often the data did simply not confirm the hypothesis put forward in the paper.

The authors are advocates of free software. But this is not the only way of guaranteeing transparency in data processing. I don't mind excel spreadsheets, if they are well documented. Often, it is absolutely sufficient to put the equations into an appendix
or the supporting informations, so that everybody can re-program the steps if wanted. Maybe more important, as a model-affine researcher I personally love matlab scripts to process data, but practically none of my colleagues in environmental chemistry, biogeochemistry, or geomicrobiology would ever do that, simply because programing is not part of their research. The approach suggested by the authors would be doomed to fail if you tried to impose it on the water-quality part of the hydrological community. For these colleagues, the quest of transparency requires different tools such as easy-to-use, flexible and yet standardized electronic lab journals and templates for excel spreadsheets linking raw data to data needed for calibration and meta information. In practice, the true intransparency lies not in handling large data streams, because people having to handle them are fully aware of data-handling and -documentation issues. The true intransparency lies in the type of research where all data still can be stored on the PC of an individual PhD student. Some funding organizations and publishers have formulated policies that data need to be stored in a repository and made be available to the public. However, for many people it's still an enigma how exactly this should be done. If there was an industry standard sold by microsoft for the management of lab data, I would not be happy that a private company makes all the money, but I would have hope that data won't simply vanish once the responsible doctoral student has left.

In contrast to the authors, I can understand strong resistance against making all source codes freely available. A specific argument against open-sorce codes relates to intellectual property rights. The development of the MODFLOW family of codes has been funded by USGS with the obligation to provide the source code. FEFLOW, by contrast has been developed by a company that has invested thousands of work hours into the development, and the return of investment relies on licencing the code. If you don't get a public-budget salary (like professors and federal administrators do), you have to sell the product of your work. If codes have been scrutinized enough by benchamrk tests etc. (which costs human resources, too), the users can rely on them without having access to the source code. So the transparency would lie in providing input files to

certified codes rather than providing the source codes.

A second reason for hiding source codes is quality assurance. Everybody can modify an open-source code, which can lead to erroneous behavior of the code. Who is ensuring that the wrongly-modified versions of the code are not disseminated? So-called community codes (like the community land-surface model - CLM, used in weather and climate modeling) require an institution that is willing to manage the official releases of new versions and perform the benchmarks. Somebody has to pay for that. The authors may believe to have reached paradise once every code is free. While I personally doubt that, it particularly is not a prerequisite for transparency in handling scientific data. So please, don't mix that up.

Let me come to my last ugly remark. If all steps of data processing was made publicly available, would the information actually be read? And - since the extra information would now be part of the publication - who would be willing to review it? As an associate editor of Water Resources Research, I once checked a matlab code provided by authors as supporting information, and I actually found an error. While on a psychologic level this gave me the illusionary ego-boost of feeling superior over the authors of that particular paper, I doubt that I would be doing that for all papers that I review or handle. Hence, the transparency by open codes requested by Mike Fienen and Marc Bakker would not guarantee that publication of erroneous analyses was prevented.

---

## Referee Comment (RC2) · W. Nowak (Referee) · 3 Jun 2016

This discussion voices an overdue opinion that should be heard by many scientists (within and outside hydology): the entire path that we take from data over analysis, modelling, simulation and so forth up to our final conclusions must be laid open, so that it can be reproduced by others - only this way we can offer transparency and falsifiabilty of research results within and across groups and in later years.

A previous comment perceived the article as making statements only about "open source codes", but I do not share that impression. The main gist of the opinion given in the manuscript is that one should "use scripts instead of manual procedures, because than everything is at least - technically - repeatable, and others can then test whether

they find it reproducible". Only when one has such scripts that denote all steps taken from initial data import (over model building,....) to final data analysis routines and their result plots from which the conclusions are drawn, then the data/computer part of our work becomes repeatable. Such scripts will then have to be published as open source. I did NOT understand the article as "just use open source codes and you are fine".

I suggest the authors re-consider whether the text may mislead some reasders to think that one could simply "use open source and you are fine".

I fully support the opinion of "script what you do and then make the scripts (and all softwares used) available together with the original data".

The only other recommendation I have is the following: there is more about "rules for proper scientific conduct that aim at transparency and repeatability" than what is said here. I think, however, that the authors do not want to give a full review of all possible and required measures in general, but they rather want to provide an opinion on the specific "data/analysis/computer" part of scientific work. For example, there is "open statement of assumptions", "clear/honest referencing", "publication of negative results", and so forth. The authors should add a disclaimer to make sure they are not being misunderstood.

With these two suggestions for improvement, I would fully support the publication of this material, so suggesting a minor revision.

---

## Author Comment (AC1) · 6 Jun 2016

We appreciate the response from Dr. Wolfgang Nowak on our Opinion Paper. We are glad that he interpreted our intent to highlight the value of a documented path from original data, through analysis and modeling, to forecasts or model results. We appreciate that Dr. Nowak recognized we were not simply implying open-source software was the answer. Nonetheless, in response to the other review, and at the suggestion of Dr. Nowak, we will revise the paper to make that clearer and hopefully avoid the misunderstanding of our conclusions as "use open source and all is fine."

The second recommendation from Dr. Nowak was to disclaim the fact that we are addressing only one issue (data provenance and auditable pathways through data and

analysis) but there are others that can enhance transparency. This is a good point, and we will revise the paper to incorporate a bit more context in that way.

---

## Author Comment (AC2) · 6 Jun 2016

Michael N. Fienen and Mark Bakker

mnfienen@usgs.gov

We are pleased to have the detailed review from Dr. Olaf Cirpka of our opinion piece. Dr. Cirpka raises some important issues. There are a couple issues that warrant revision of our text to clarify our intent and meaning, while there are a few others we do not agree with. We have distilled Dr. Cirpka's extensive comments to a few salient issues which we address in turn.

1. *"The authors take a recent scandal ... as an opportunity to call for equally strict rules" requiring scripting of data analysis and modeling in Hydrology as was done by the Institute of Medicine:* Indeed, we suggest that repeatability is as much an

issue in hydrology, but we do not suggest to define a set of rules that need to be strictly enforced. Rather, we suggest that the techniques mandated by IOM have relevance to hydrological science (and many other fields).

2. *"discuss counter-arguments and obstacles":* As an Opinion Piece, we prefer to rely on the online Interactive Discussion (such as this one) to form the other side of the discussion.

3. *"rethink whether repeatability is really more important than reproducibility":* This is an important topic and we regret that Dr. Cirpka interpreted our brevity on reproducibility as dismissal of its importance. We will expand the paragraph distinguishing repeatability from reproducibility to better explain why our focus is soundly on repeatability in this piece. Most hydrological studies are not fully controlled experiments, as in many other fields, but take place in the field, which means that results strongly depend on the specific field site and the specific circumstances (temperature, rainfall, river discharge, etc.), which cannot be fully reproduced. Only a few hydrological experiments are truly reproducible in the classical sense, for example at the Borden and MADE sites mentioned by the reviewer. In contrast, many hydrologists are trying to understand and predict natural systems through measurement and modeling rather than performing controlled experiments that can be reproduced by colleagues.

4. *"broaden their perspective to reach out to hydrological researchers who are not modelers":* The paper extensively discusses both data analysis and modeling, and our opinion equally holds for projects that do data analysis without modeling. But, in a sense, we are all modelers. Experimental campaigns are commonly followed by data analysis, which includes some kind of modeling, even if it is "just" trying to determine a trend.

5. *"separate issues of transparency from the call of open-source programing":* We are (as our opinion) advocates of open-source programming. However, our focus

is meant to be on the techniques of recording steps taken in analysis than on open-source software particularly. But, as we state in the paper, the two are related: "Without the availability of an executable code, the simulations can still not be repeated and without the availability of the code itself, the computational steps in the code cannot be understood and scrutinized." Nonetheless, we have tried to make it clearer that open-source programming is a factor but not the entire answer.

6. *"it gives the erroneous impression that omics was specific to cancer research":* We regret that Dr. Cirpka got this impression, but we feel the context is clear and indeed it was omics that was at issue in the cancer scandal. We propose to highlight that we are speaking of omics in the context of cancer research by adding words in bold type in the sentence "The fields of omics **as used in cancer research** and hydrology may seem as completely unrelated..." which we assume was the source of Dr. Cirpka's objection.

7. *Objection to the analogy between hydrology and life science on two accords. a) physical scientists have more connection to their raw data so the repeatability steps may be applied in hydrology, and b) there is less pressure to twist results in hydrology as lives are not at stake in the same way they are in pharmaceutical research.* On the first accord, whether hydrologists have more connection to their raw data is a judgement call that we are not able to make, but we agree that repeatability steps are equally valid for hydrology (and other sciences). On the second accord, we often hear similar arguments that hydrologic findings have less at stake (at least in the short term) than national security, health, etc. While this is true, we don't agree that such a case serves as an excuse for us (as hydrologists) to be less robust in conducting our science. If we wish to take ourselves seriously (and be taken seriously by others) we need to hold ourselves to a high standard. However, and harkening back to point 1, we are not advocating for strictly enforced rules, but rather suggesting best practices that, ones adopted

by the community, will form a standard that others want to comply with as best practices.

8. *"I agree with the authors that transparency in data processing is mandatory. But this may be achieved by other measures than enforcing everybody to use open-source codes written in a free-of-charge scripting language.":* We will revise our language to make it clear that free-of-charge is not the principal criterion we are advocating, although it makes it obviously a lot easier to repeat a modeling effort when the code is free. However, Dr. Cirpka also discusses that he is bored reading about why certain data are selected and others rejected in papers, but he adds: "...much worse would be dropping the three time series and all outliers altogether and pretending that only 12 surprisingly consistent time series were taken." We agree completely! That's why scripts are much better than spreadsheets. In a spreadsheet, often one simply deletes data that are not carried forward in analysis, but in a script that operates on data with auditable provenance, every such decision to drop a member of a dataset (or perhaps make a judgement call about data quality, such as a unit or datum conversion) can be documented.

9. *"The authors are advocates of free software. But this is not the only way of guaranteeing transparency in data processing. I don't mind excel spreadsheets, if they are well documented.":* The paragraph following this comment has a fair bit to unpack. First of all, Excel spreadsheets certainly play a role, but with complicated data analysis, often the order of operations and details about the calculations is difficult to fit properly in the confines of a spreadsheet. We advocate scripting languages in this piece, but even Fortran or C source code can contain detailed comments that might make it easier to follow the progression of calculations made to process data. RScript, MATLAB, and Python notebooks are much better than that. We also strongly disagree that placing the equations in an appendix is "absolutely sufficient." It is one thing to work out equations properly

and entirely another to actually implement them correctly. Many errors are not as egregious as using the wrong equation, but are things like a unit conversion not made (that's how to crash a spaceship into Mars!) or even a wrong sign. The scripts contain not only notes but the actual calculations and since the scripts have been written anyway, why not make them available? We totally agree that archiving data in a meaningful way is a challenge, but that's a weak excuse not to try. Forcing PhD students to be more transparent and create repeatable work would be a service to the community going forward. Finally, it's disappointing to hear Dr. Cirpka thinks that enforcing such requirements on the water-quality community is doomed to fail. Indeed, the omics research related to cancer research that motivated this piece has more in common with water-quality than with quantitative hydrology. It seems like an excuse not to try rather than a solution.

10. *"On free software":* Just three additional points on free software to address here. First, to imply that because a code has been paid for it is bug-free is naive - patches are constantly made to commercial software partly because bugs are found by users. The inability for a user to inspect the code is an impediment to quality assurance. Sure, not many users will want to look at the code, but enough do so that it can enhance quality. That said, we do not mean to imply that all software must be free. FEFLOW is a solid code, as are many in the petroleum industry which are expensive and commercial. But, it's simply not true that "If codes have been scrutinized enough by benchamrk*[sic]* tests etc. (which costs human resources, too), the users can rely on them without having access to the source code." Scrutiny and analysis by users is key to any software development and hidden bugs in proprietary code can go unnoticed much longer than when external scrutiny is ongoing. Second, open-source and free software is not the same as unmoderated community software. All three exist in various combinations, but many open-source projects are maintained by a team (who often sell training and consulting services rather than shrinkwrapped boxes of software to

pay the bills). In fact, many open-source codes can be purchased so are not technically free. Many tools have been developed for open software development to enforce rigorous testing and quality checking by a limited team. Git makes this available through putting the code online so anyone can modify their own copy, but the lead developers decide which proposed changes get accepted.

11. *Finally what the reviewer called an "ugly" comment on whether anyone will read the data analysis scripts:* It is not necessary that all the scripts be reviewed by journal editors and peer reviewers. It is fine in our view to assume that the calculations are correct while reviewing a paper. But...having such scripts and notes available to the reviewer can be valuable when results don't make sense or an error is suspected. They need not be read and scrutinized in every case to be valuable. Certainly many cases have little at stake and will not be reviewed, but in some cases the audit may be crucial.

---

## Referee Comment (RC3) · W. Nowak (Referee) · 7 Jun 2016

With pleasure I see the discussion evolve. I have read the response of the authors to the comments by Dr. Cirpka - there is a line of discourse about the question whether data processing scripts provided as supplementary material are a burden or a blessing during review. While I fully agree that one cannot expect reviewers or Associate Editors to suddenly look into a much larger volume of details, I agree with the authors reply that - in case of doubt - such additional scipts may be worth looking at.

However, there is an additional point here: How many times have authors been contacted five or more years later with requests like "Nice work you did back then, I would like to use your procedure to follow up on that study (transfer of a well-done analysis to

a different study region; use of the developed methods; upgrade of the methods developed)". Or "I am doing follow-up research (same/different study region; application of methods; improved methods,... ), and I have problems in repeating/reproducing your results". Having everythink available in a repeatable manner has two advantages: in case an error is detected, the follow-up research activity can falsify the previous results (ending in an erratum or a comment/reply series in the journal, both of which would help to assure quality); in case there is no error, the follow-up study confirms and then runs faster because past efforts in analysis/coding can be recycled. The former is about repeatability for testing and falisification, the latter is about re-use and citation boost for one's own ideas.

Well, this is a slight side track as fifty percent of it is about re-use, speed and citation boost of one's own scientific results. However, the other fifty percent are about classical repeatability by others - outside the peer review process but within the years to come. If we look back at the evolution of science, then the largest steps were achieved by falsifying older theories, and this did not happen during peer review but in later years after publication.

One might think of the "marketing" aspect as an additional incentive to persuade researchers to follow repeatable concepts, and this may come to the rescue when questioning whether the hydrology community can ever be persuaded to enforce repeatability.

The authors might take this side track discussion to add an additional paragraph to their opinion if they deem it useful for their manuscript. I would be happy to see this happening, because I have hope that it can indeed help to persuade (at least some hydrologists) .

---

## Referee Comment (RC4) · A Bellin (Referee) · 11 Jun 2016

This opinion paper addresses a very important and sensitive aspect of documenting research results with an emphasis on hydrology. The authors move from a short description (too short indeed) of the Duke cancer research scandal, introduce the concept of reproducibility and repeatability and discuss how the latter applies to hydrological studies. After observing that in hydrology like in "omics" (the science involved in the Duke scandal) some type of modeling is applied to forecast the behavior of the analyzed system and that both use large datasets, which need to be processed, trimmed and validated before using them in modeling, the authors conclude on the need for the hydrological community to organize modeling works such as to guarantee full repro-

ducibility of their simulations.

The opinion is well written and addresses an important issue, which is currently debated in the hydrological community. However, given the topic and the context I was expecting more comprehensive conclusions and suggestions on how to implement this concept in hydrological sciences. Adopting open source tools and collaborative coding environments is highly recommendable, but it is only an aspect of a more complex picture involving data sharing and methodological developments, including the correct quantification of uncertainty, which has not been considered in the opinion. In the presentation the authors mix technological choices (such as scripting instead of GUI) with the more general issue of reporting modeling efforts with a level of detail enough to allow the reader to repeat all the computations. I believe the second aspect is much more relevant than the first one and requires a reversing of the actual trend to publish in "letters" form. In addition, technology is today available for data and modeling sharing, to a higher level than we currently do. This of course requires a change of perspective, moving from a competitive type of environment to a more collaborative one, in which anyone can build more easily than today on the results of others. A technological solution helping in reporting, within a script, all the steps of the modeling effort is important but only an aspect of the entire picture. I suggest to separate these two points and comment more on the second aspect of the modeling effort. Furthermore, there are still cases in hydrology in which data are scarce, a situation that is in sharp contrast with the "omics", in which large datasets are available and some trimming is always necessary. In hydrology trimming is a luxury that in many cases we do not enjoy. In hydrology the concept of repeatability is broader than described by the authors, since it is recognized that different models, with the same dignity, may lead to different results (epistemic uncertainty) and the same model can produce different, yet likely, results depending on the chosen parameters (parametric uncertainty). This is common in all natural sciences, indeed. The strict definition of repeatability requires that the same model with the same set of parameters is run a second time by another independent group to check if the same results are obtained, but this does not guarantee about correctness of the interpretation and conclusions of the study. It is "just" a technical verification of the modeling exercise. How can be repeatability defined in a context in which we recognize that uncertainty plagues our modeling efforts? This is a question, which I think may deserve space in this opinion paper. However, giving the nature of the paper this is not a request to revise, in case of a different opinion by the authors. Overall, the manuscript is well written and it may be instrumental to stimulate discussions in the hydrological community on this relevant topic.

---

## Author Comment (AC3) · 11 Jun 2016

It is good to see a robust discussion developing around this opinion piece. Dr. Nowak agreed with our point that archiving data and processing through scripting is worthwhile even if not everyone will examine every step. We are glad he sees this point.

Dr. Nowak also raises another point that repeatability transcends a need for transparency and is useful for future researchers to revisit work done in the past. This is an excellent point! We respond here with a bit of discussion but will also briefly highlight that issue in the revised manuscript. In this context, however, we would like to reply more substantially.

[Figure]

**When the public pays for research** At the US Geological Survey, where the first author is employed, there are policies in place in groundwater hydrology (and rapidly expanding to other research) to maintain rigorous archives of models and data analysis. These archives are designed to make it possible for the taxpayers to obtain copies of models that are operational and match the results published in papers and USGS technical reports. These archives require a fair bit of work to assemble, require a peer review which takes time and energy, and many languish for years without any interest—until there *is* interest. Recent projects have focused on automatically scraping the archives to assemble, for example, all published USGS models in a large region of the United States. Without maintaining the archive, such new meta-projects would be impossible. Many consultants and researchers also, upon learning that USGS has created a model of an area they are interested in, request the archive and thus have a working model and (as we stated in the manuscript, possibly of more interest) the supporting data to launch their new project from. Using a scripting approach to also make the steps of data processing and analysis available serves two purposes: it makes the process of archiving easier and more transparent, and it provides the context of interpretations made by the original researchers as they evaluated the data and made the model.

**Marketing** We also agree that trying to enforce more rules and extra work is likely to be met with skepticism and disdain (indeed, the authors have experienced that both in conversations with colleagues and in discussion in this forum). However, Dr. Nowak makes a good point that even voluntary standards and protocols are of value if researchers can market their work as following them. Consumers of the results–be they other researchers, consultants, or the public–can demand higher standards even as they remain voluntary.

It is true that we are wandering into a different topic than the original intent of the piece, but these issues are useful and important so we will make mention of them briefly in

the revisions to the manuscript.

---

## Referee Comment (RC5) · S. Geiger (Referee) · 13 Jun 2016

The paper by Fienen and Bakker is an important opinion piece that argues that data analysis, and more generally scientific findings, in hydrology need to be both, repeatable and reproducible (providing a careful distinction between the two terms), to avoid future scientific scandals such as the infamous Duke cancer study. The authors view scripting languages, published communally through an Open Source approach, as a key enabling technology that can help to support repeatable and reproducible science. They further argue that academia needs to develop a proper reward structure where the publications of well-tested and carefully documented code is equally recognised as the publication of a peer-reviewed paper.

I think that this opinion piece is a timely contribution to HESS, and part of a larger groundswell that calls for repeatable and reproducible science across the entire research field of porous media flow. For example, a forthcoming editorial statement in the SPE Journal will make similar arguments for repeatable and reproducible science in the field of petroleum engineering.

I think this opinion piece should definitely be published but would welcome if the authors take a more differentiated view that also addresses some of the wider challenges related to generating repeatable and reproducible science by considering (some of) the following points:

1. While the Duke Cancer study is one "good" example of a high-profile scientific scandal, there are unfortunately other high-profile examples where experiments, data, and analysis were not properly documented and publications selectively used the results to satisfy foregone conclusions, and in some cases even fabricated the results. The largest scientific fraud in the field of physics involving the German wunderkind Jan Hendrik Schoen at Bell Laboratories at the turn of the millennium springs to mind. Improper use of data is a much wider issue that goes beyond the field of "omics", although the consequences may be very different: Cancer research directly impacts human life and inappropriate use of data could be a life-or-death decision; the Schoen scandal involved a research field were scientific breakthroughs could have been rewarded with a Nobel Prize. Hydrology, or more generally porous media research, normally does not have this kind of impact, for better or worse. Where the stakes are high (e.g. when simulating the performance of hydrocarbon reservoirs to help supporting drilling multi-million dollar wells), stakeholders normally have strict protocols in place that aim to assess the quality of the work and mitigate risks. However, what all scientific scandals have in common is that they adversely impact the scientific careers of innocent bystanders, especially the careers of PhD students and postdocs who repeatedly and unsuccessfully try to reproduce the fraudulent results as a basis for starting their own research.

[Figure]

2. Scripting and Open Source code development is just one means to reach the goal of reproducible and repeatable science. Equally important is the stewardship of the underlying data, which may have been collected as part of a larger interdisciplinary study and is published along with the analysis. The Research Councils UK (RCUK) are now enforcing that UK universities develop policies that guarantee that all data from all publicly funded research appropriately managed and archived at the university that originated it (for example, see the guidelines for the field of engineering and physical sciences at https://www.epsrc.ac.uk/files/aboutus/standards/clarificationsofexpectationsresearchdatamanagement/). Only by making the underlying (experimental and/or field) data available as well as the analytical tools (scripts and/or code) available, science will become truly reproducible and repeatable. An important question, however, then becomes who will manage the data; as we are currently finding in the UK, this is often not an easy and cheap task to accomplish. A repository like GIT may provide a convenient solution, but who will manage the repository once the PhD student or postdoc who wrote the scripts has moved on or the funding has ceased? And how should we store gigabytes or even terrabytes of raw data?

3. I do not believe that the use of GUIs or, indeed, the use of commercial software is the key problem when it comes to reproducible and repeatable science. As it should be common with laboratory experiments, simulation experiments should also maintain a "lab book" that documents how certain simulation experiments were run (e.g. which input parameters were used in a simulation as well as the decision making process that has led to the particular choice of parameters, which may well include a large number of "failed" simulation attempts – but see the above comments on how to manage potentially vast quantities of data) . Such metadata may be documented directly in the script itself or electronically (including the version of the software that was used). I would further add that the generation of data, be it in the lab or field, can be the main focus of a study and the analysis is only a minor part of the research that relies on existing software packages with an easy-to-use GUI; not every student will become an expert

in experimental research AND scripting languages.

4. Although I am a big advocate of Open Source code, practice often shows that Open Source code is not as well documented as it could, and that a simulation with certain parameters that was running in, say, version 1.1 no longer runs in version 1.2. The nature of this is perfectly understandable: The code is often developed by students and postdocs who have other targets than documenting and testing the code to the level where it can be readily run by a large number of scientists. The authors rightly state that academia needs to change its reward system to recognise significant code developments, but in the current highly competitive "publish or perish" climate I am not overly optimistic that this change will happen anytime soon. To this end, I would hence welcome if the authors actually discuss some examples of best practice in Open Source code development such as the Open Porous Media initiative, and herein in particular the Matlab Reservoir Simulation Toolbox, where all input scripts and input datasets are provided along with the scientific paper (see http://opm-project.org/).

5. A possibly contentious issue surrounding the provision of Open Source code is plagiarism: Where code is made available at the time of publication, it becomes much easier to repeat an analysis – perhaps with a negligibly small change – and pass the "research" off in another (perhaps less well-read) journal as your own work. As stated in the editorial of Water Resources Research a few years ago, and as I have witnessed myself as the associate editor for two journals, plagiarism is on rise. In theory, publishing code and data along with the paper provides could offer evidence as to where the original research was conducted, but still plagiarism it is probably the most frequent counter argument that I have heard when it comes to making code available via an open source route.

---

## Author Comment (AC4) · 13 Jun 2016

We thank Dr. Geiger for his support of our Opinion Paper and for his suggestion on how to improve it. Dr. Geiger raises the following five points:

**Other scandals** There are unfortunately many other scandals on scientific studies where things went awry. The Netherlands (the home country of the second author), the scientific community was shocked in the past few years by a high-profile social scientist that had made up all kinds of data, which indeed had detrimental effects on his PhD students and the credibility of the scientific community. Every country seems to have its own high-profile cases, but many of these cases

concern deliberate activities to falsify data. Such cases are very difficult to catch if done 'well' and are not the topic of our Opinion paper, and we will revise our paper to indicate that. We have built our Opinion Paper around the Duke Cancer Research scandal, as this is a prime example where the researchers did not deliberately falsify data, but published results that could not be repeated as they were based on a few questionable choices. The protocols developed by the Institute of Medicine are intended to make it possible to repeat published results, and we were inspired to address similar practices in the field of Hydrology.

**Stewardship of underlying data** The issue of how to store and make data and code available is an important one. In the United States, this has been recognized at the highest level (see White House memorandum referenced in the Opinion Piece). It can certainly be cumbersome to follow rigorous data-handling protocols with large datasets but the reward is large in the future and there is an obligation for the public to have access to data that society pays for. Our main point in this work, however, picks up where the data stewardship leaves off. Documenting the path from original data through analysis and potentially forecasts is the context in which we write.

**GUIs and commercial codes vs. scripts** One of the main points of our paper is that research is not repeatable when a GUI is used, unless every button-push (and the order) are recorded. Our suggestion was to record such button-pushes in a script. As mentioned in the paper, several GUIs already have this capability, which makes the analysis instantly repeatable. Such a 'spit out a script' option will make it possible for researchers to produce repeatable research without becoming scripting experts.

**Open Source** The documentation of Open Source codes is indeed an issue. Writing documentation is considered (at least somewhat) boring by many code developers (including the authors of this paper), but, obviously, crucial. In that respect
we will emphasize this when discussing that the development of codes *and documentation* needs to be rewarded more appropriately by academia. We will think about if we can add a discussion of examples of best practices to make source codes available.

**Plagiarism** We are aware that some researchers don't want to make their code available, because they don't want others to change it a bit, then use it on their own problems, and then publish it. Luckily this can be regulated with the choice of an appropriate Open Source license, which gives authors the ability to specify what can and can not be done with their code. Further, the more detail of work is documented, the easier plagiarism can potentially be detected. It is indeed good to mention these issues in our paper.

---

## Author Comment (AC5) · 13 Jun 2016

We thank Dr. Bellin for contributing to the discussion about our opinion piece. We are pleased that Dr. Bellin generally found our comments of interest and we welcome the suggestions for potential improvement of our presentation.

**Considering uncertainty** We are definitely advocates of considering uncertainty in all scientific endeavors. However, we have the opinion that discussion of uncertainty is really a topic in itself and not closely enough related to our main topic to add extensive discussion of it in the opinion piece. However, we will make strides to clarify (also in response to other comments) that repeatability and reproducibility

are different. What Dr. Bellin identifies as a shortcoming by not accounting for the role of parameter and epistemic uncertainty really points to the difference between the two. Being able to repeat analysis and modeling with a specific parameter set may seem trivial since it doesn't consider the uncertainty of the process, but it is a necessary and often difficult to accomplish step! Even stochastic approaches should be repeatable and ensembles of parameter sets and resulting forecasts can be carried forward using scripts. This does not guarantee reproducibility in the case of epistemic uncertainty as it influences subjective decisions about the data/models, as another group with another model may come to different conclusions. If this is the case, it is crucial that the published results can be repeated, as it can at least be concluded that the difference is not due to errors in the published results but (likely) due to epistemic uncertainty. This highlights again that reproducibility is still an important issue that is not given enough attention in the hydrological sciences. We will clarify this in the revised version of our paper and highlight the importance of repeatability in the case of epistemic uncertainty.

**The more general issue is to include more details than "letters" style articles** Indeed, there are multiple aspects to how more detailed information leading to greater repeatability can be incorporated into scientific discourse. This was also an issue raised by Dr. Cirpka. However, we chose to highlight the response of the medical research community to the Duke Cancer Research scandal, being to require not just detail in writing, but an executable path through analysis via scripting. We use this as an example rather than insisting on this as the only solution. We are glad that the result has been a vigorous discussion so far and we will incorporate more about the general issue of repeatability in the revised manuscript.

**Scarcity of hydrologic data** It is true that the example from omics often are cases with large datasets that must be trimmed while in hydrology data are often scarce so trimming is less an issue. We can clarify this in our paper. However, the analogy between the fields is more basic in our view. Whether the issue is trimming a large omics dataset or interpreting noisy and sparse hydrologic data, in both cases subjective decisions must be made about suitability of data. Since they are subjective, other researchers must be able to understand, assess, and, possibly, overrule such interpretations. By clearly documenting them in a scripted path through the analysis, other researchers can change, add, or subtract their interpretations of the data and rerun the analysis. Such transparency can also enhance the level of collaboration Dr. Bellin hopes for. Using tools that are freely available further enhances that ability.